# An ancient river landscape preserved beneath the East Antarctic Ice Sheet

Stewart S. R. Jamieson [1,7] ✉, Neil Ross [2,7], Guy J. G. Paxman[1], Fiona J. Clubb [1], Duncan A. Young[3], Shuai Yan[3,4], Jamin Greenbaum [5], Donald D. Blankenship [3] & Martin J. Siegert[6]

The East Antarctic Ice Sheet (EAIS) has its origins ca. 34 million years ago. Since then, the impact of climate change and past fluctuations in the EAIS margin has been reflected in periods of extensive vs. restricted ice cover and the modification of much of the Antarctic landscape. Resolving processes of landscape evolution is therefore critical for establishing ice sheet history, but it is rare to find unmodified landscapes that record past ice conditions. Here, we discover an extensive relic pre-glacial landscape preserved beneath the central EAIS despite millions of years of ice cover. The landscape was formed by rivers prior to ice sheet build-up but later modified by local glaciation before being dissected by outlet glaciers at the margin of a restricted ice sheet. Preservation of the relic surfaces indicates an absence of significant warm-based ice throughout their history, suggesting any transitions between restricted and expanded ice were rapid.

The glaciation of Antarctica was triggered by global climatic cooling over the Cenozoic Era (Fig. 1). During the Eocene, glaciation was likely restricted to ephemeral ice masses and small-scale mountain glaciers in regions of high topography[1,2]. However, a step-change in ice extent and volume occurred at the Eocene-Oligocene transition (EOT; ca. 34.0–33.5 Ma; Fig. 1) when the first widespread Antarctic glaciation was recorded in marine sediment records[3,4]. This transition to a glaciated Antarctica was potentially caused by a combination of $CO_2$ dropping below a key threshold[5], associated feedbacks within the carbon cycle[6,7], and the opening/deepening of circum-Antarctic ocean gateways[8]. In East Antarctica, the ice sheet likely nucleated on the high topography of the Gamburtsev Subglacial Mountains, Transantarctic Mountains, and Dronning Maud Land[2,5,9] (Fig. 1). Expansion and coalescence of independent ice masses on these highlands led to the growth of the continental-scale EAIS[5,10].

The EAIS has fluctuated on a range of spatial and temporal scales since the EOT, with warm-based glaciers reaching the coast and advancing and retreating at orbitally-paced timescales during the Oligocene[11,12], and major variations in ice extent and volume occurring during the early- to mid-Miocene[13–16]. Following the Mid-Miocene Climatic Optimum (ca. 17.0–14.8 Ma; Fig. 1), further atmospheric cooling led to the establishment of an arid polar climate, and by ca. 14 Ma, a continental-scale ice sheet became persistent, with evidence for widespread cold-based ice and advances and retreats of the ice-sheet margin to and from the continental shelf edge[17]. Offshore sedimentary records suggest the EAIS may have experienced retreat during subsequent periods of warmer climate, for example, the mid-Piacenzian warm period (mPWP; ca. 3.26–3.02 Ma)[18–20] and potentially the Pleistocene interglacials of MIS 11c (ca. 430–400 ka)[21] and 5e (ca. 125 ka)[22–24]. Numerical ice sheet models indicate that retreat may have been focussed in low-lying marine-based sectors of the EAIS, such as the Wilkes and Aurora Subglacial Basins (Fig. 1) during Plio-Pleistocene warm intervals, although the extent of this retreat remains uncertain[25–27].

Much of the evidence for past ice margin fluctuations and dynamic change of the EAIS is derived from global records of sea-level

[1]Department of Geography, Durham University, Durham DH1 3LE, UK. [2]School of Geography, Politics and Sociology, Newcastle University, Newcastle upon Tyne NE1 7RU, UK. [3]University of Texas Institute for Geophysics, Jackson School of Geosciences, University of Texas at Austin, Austin, USA. [4]Department of Geosciences, Jackson School of Geosciences, University of Texas at Austin, Austin, USA. [5]Scripps Institute for Oceanography, University of California at San Diego, San Diego, USA. [6]Tremough House, University of Exeter, Penryn Campus, Penryn, Cornwall TR10 9FE, UK. [7]These authors contributed equally: Stewart S. R. Jamieson, Neil Ross. ✉e-mail: Stewart.Jamieson@durham.ac.uk

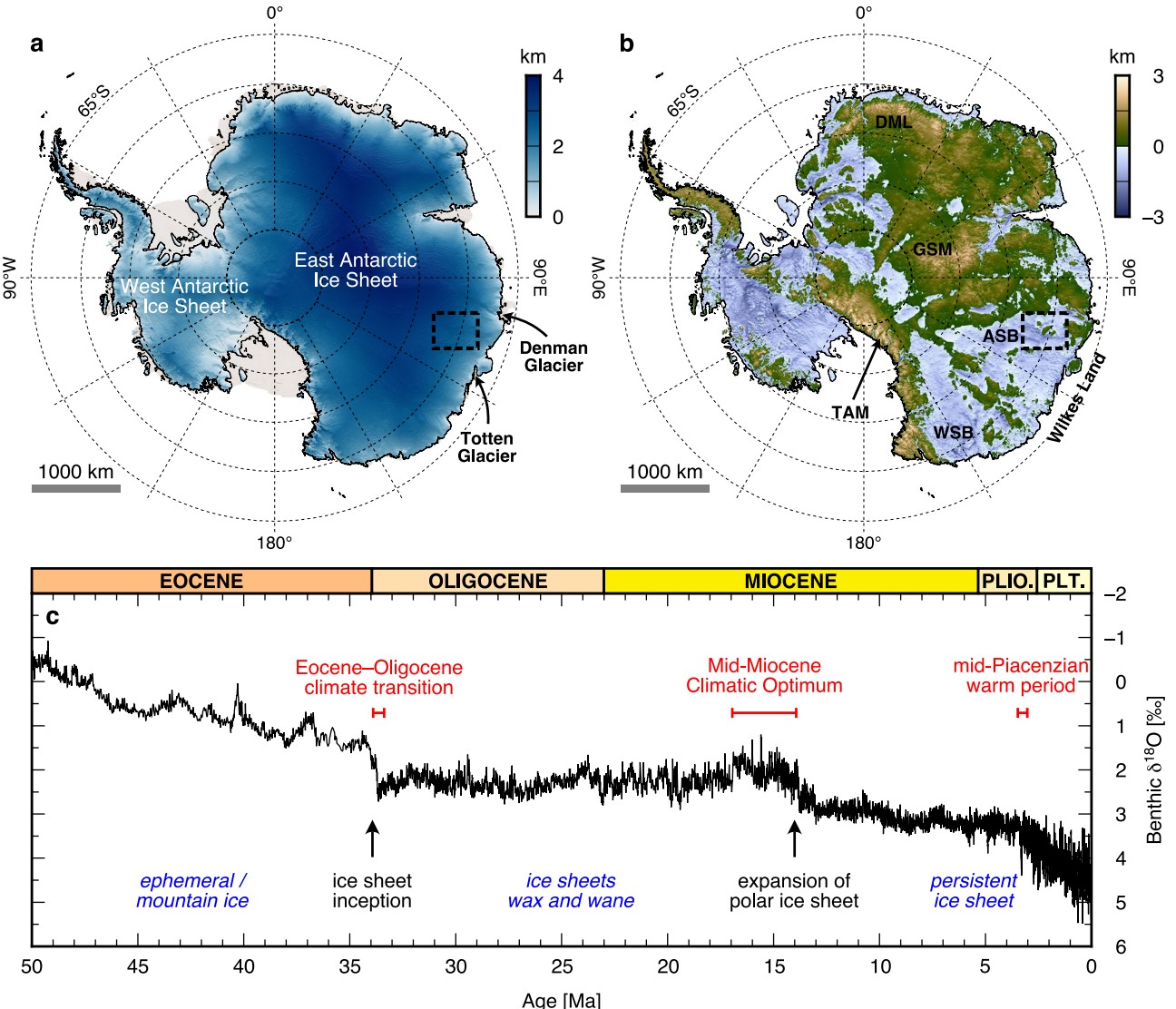

**Fig. 1 | Location and climate evolution. a** Surface elevation of the Antarctic Ice Sheet from the Reference Elevation Model of Antarctica[68]. The black line marks the present-day grounding line[74]; pale areas are floating ice shelves. **b** Bed elevation of Antarctica beneath the grounded ice sheet from BedMachine v.3[64]. Surface and bed elevations are relative to the global mean sea level (the EIGEN-6C4 geoid). The black dashed box shows the extent of the maps in Figs. 2 and 5, and lines of latitude are separated by 5°. ASB Aurora Subglacial Basin, DML Dronning Maud Land, GSM Gamburtsev Subglacial Mountains, TAM Transantarctic Mountains, WSB Wilkes Subglacial Basin. **c** Composite global oxygen isotope record from benthic foraminifera[75]. Lower (more negative) $\delta^{18}O$ values indicate lower ice volume and/or warmer ocean temperatures; higher (more positive) $\delta^{18}O$ values indicate greater ice volume and/or cooler ocean temperatures. Labels show the timings of major climate intervals or transitions (red), glacial expansions (black), and prevailing glacial conditions (blue). Plio. = Pliocene; Plt. = Pleistocene; note that the Holocene epoch (0.012–0 Ma) is not shown on the timescale.

change and/or ice volume[28] or from local records of glacier fluctuation on the exposed land around the coast and in the offshore sediments on the continental shelf[12,18,19,29]. Although we have been able to infer from these lines of evidence that significant past changes in the EAIS have occurred, there is limited direct evidence that records the degree to which the ice margin retreated inland of its present-day position during past warm periods. This means the extent and possible rates of past EAIS retreats are poorly resolved. However, as ice sheets fluctuate, they modify the landscape upon which they rest, leaving a fingerprint representing the long-term average glacial conditions to which it has been subjected[30]. Interpretation of subglacial topography, measured using radio-echo sounding (RES) surveying, can therefore be used to infer past processes of landscape evolution and former ice dynamics[9,31–35]. Processes that may be reflected in the subglacial landscape include but are not limited to, glacial erosion, solid earth adjustments to erosional unloading, and fluvial incision under

scenarios of more restricted EAIS extent[36]. Accordingly, interpreting the morphology of the East Antarctic subglacial landscape and its long-term evolution offers a significant and hitherto relatively little exploited opportunity for understanding the past extent, fluctuations, flow, and thermal regime of the EAIS.

Here we use geophysical data to analyse an area in East Antarctica with a subglacial landscape that appears to be at odds with the modern ice sheet. It lies adjacent to the low-lying Aurora and Schmidt Subglacial Basins and inland of the Denman and Totten glaciers; a sector of the EAIS known to be sensitive to past and potentially future climate and ocean warming (Fig. 2)[13,33]. Young et al.[37] identified palaeo fjord landscapes cutting the highlands that bound these basins (Fig. 2a), interpreting them as evidence of Oligocene and/or early Miocene incision (i.e., between ca. 34 and 14 Ma) under an ice sheet significantly smaller than today[37]. Our geophysical study focuses on the higher elevation land (now bed) surfaces between the fjords, which may be

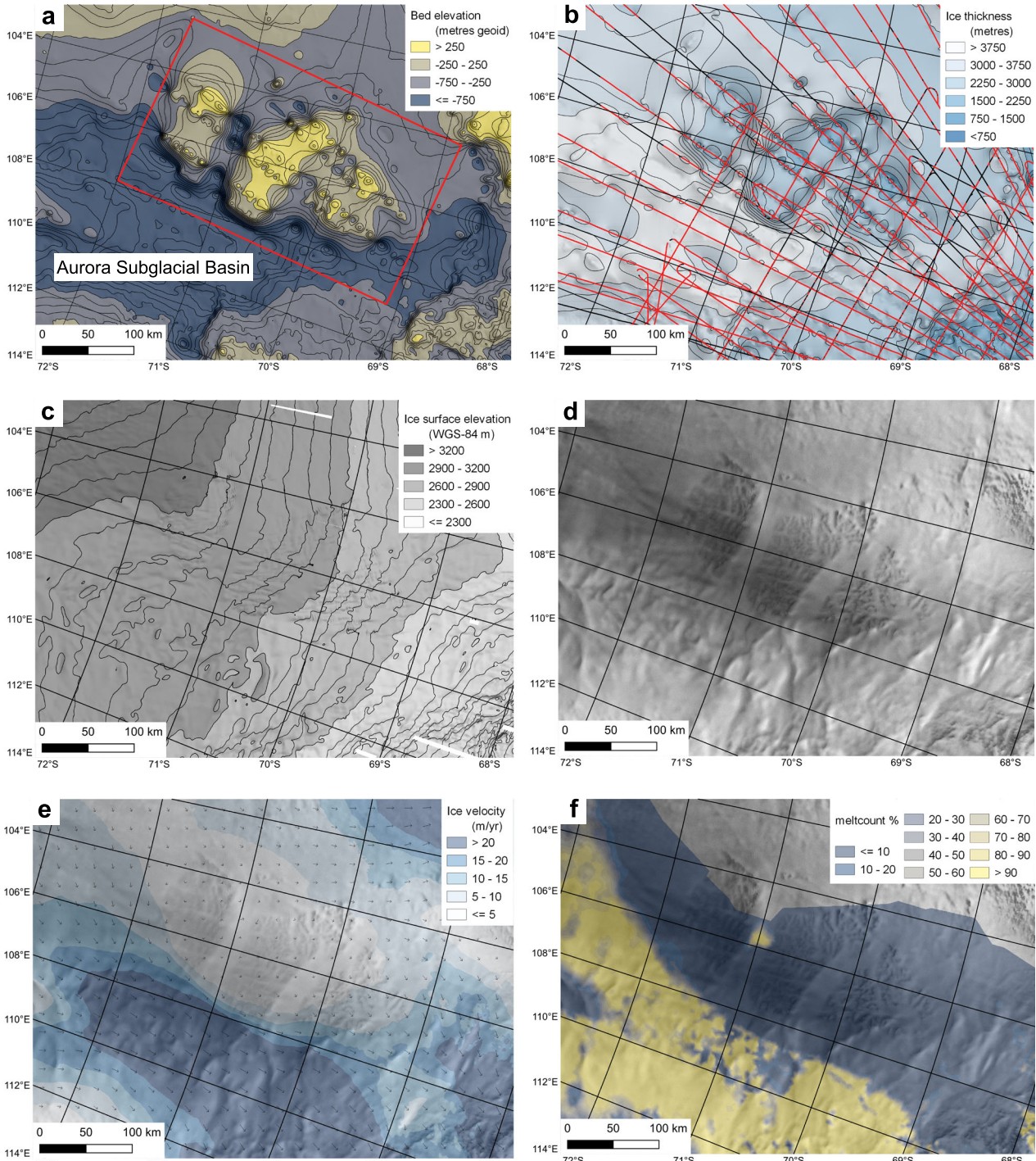

**Fig. 2 | Characteristics of the subglacial highlands and troughs of 'Highland A'[37].**
**a** BedMachine version 3 bed elevation[64], contours (black) in 250 m increments. Bed dataset is transparent and underlain by a hillshade of the same dataset. Extent of Fig. 3a shown by red box; **b** BedMachine version 3 ice thickness[64]. Black lines are radar profiles, with red parts showing where bed was picked. Ice thickness contours (black) are in 125 m increments. Ice thickness dataset is transparent and underlain by a hillshade of the same dataset; **c** Ice surface elevation[68], contours (black) in 50 m increments. Ice surface dataset is transparent and underlain by a hillshade of the same dataset; **d** RADARSAT image mosaic of Antarctic ice surface morphology[42]; **e** ice sheet velocity[41]; **f** Basal melting[47] identified as a percentage of 500 ensemble members recording basal melt. Backdrop imagery for (**e**) and (**f**) is a RADARSAT image mosaic of Antarctica[42].

more ancient features preserved beneath the EAIS. Our overarching aim is to evaluate the evolution and age of these land surfaces and to assess their implications for EAIS history.

To test the hypothesis of an ancient land surface preserved beneath the modern EAIS, we quantify landscape characteristics from RES data collected as part of the International Collaborative Exploration of the Cryosphere through Airborne Profiling (ICECAP) airborne

geophysical survey of the Aurora Subglacial Basin and Totten Glacier between 2008 and 2011[33,37–39] (Fig. 2b). To support this hypothesis, we would expect a specific set of criteria to be met, which would be relevant anywhere on the continent. These are: (1) topographic features should be similar to each other but not consistent with continental-scale ice flow; (2) the elevations of the surfaces should be coherent when corrected for more recent glacial trough incision and

ice loading; (3) the basal thermal regime must be cold-based to preserve the landscape. To evaluate these criteria, we: (1) map topographic features using satellite data that record changes in ice surface slope, which have previously been shown to reflect the large-scale subglacial topography[32,40] (Fig. 2d); (2) use along-track RES data (Fig. 2b) to ground-truth the geomorphology and quantify the subglacial topographic relief; (3) apply flexural modelling to understand whether the highland blocks are consistent with having once been a single land surface that was subsequently incised and uplifted by selective linear erosion[37]; (4) evaluate our findings in the context of modern and past ice flow and basal thermal conditions.

## Results

We identify three distinct landscape blocks that constitute 'Highland A'[37]. The blocks range between 121 and 173 km long and 73 and 85 km wide; each covers an area between ~7000 and 10,000 km² (Fig. 3). The blocks are separated by two large-scale troughs, which are up to ~40 km wide and thought to have been generated by selective glacial erosion, hence previously referred to as 'fjords'[37]. The southern trough is significantly deeper than the northern trough, with minimum elevations of 1480 m vs 620 m below global mean sea level, respectively. On a regional scale, the three highland landscape blocks appear to have similar morphological characteristics, each comprising rough terrain containing an intricate network of ridges and valleys (Fig. 3). If we were to assume these landscape blocks were once a coherent single land surface, it would have covered an area of more than 32,000 km² (~1.5 x the area of Wales) with a long-axis of ~300 km. RES data confirm that across the three highland blocks, the buried topographic peaks have relatively uniform heights of 660–850 m above global mean sea level, with a subtle 'domed' pattern of tilting away from the southern and central blocks, which exhibit the highest elevations (Fig. 3).

The first criterion, that the landscape fragments should be similar to each other but should not be consistent with modern continental-scale ice flow, was assessed by quantifying the valley morphology and comparing it to modern ice velocity data (Fig. 2e). Currently, modern ice flows broadly northwards towards the Wilkes Land coast and is very slow (<5 m yr⁻¹) over the blocks and slightly faster (~25 m yr⁻¹) as it flows over the deeper basins surrounding our study area[41]. RADARSAT ice surface imagery[42], calibrated with airborne RES data[38,39], shows that the landscape blocks host a complex network of trunk and tributary valleys separated by sinuous mountain ridges, with valleys ranging between approx. 3 and 75 km in length (Fig. 3a).

Spatial analysis shows that the valley networks on all three highland blocks have a similar structure, with two dominant, near-orthogonal trends in valley orientation (approximately E–W (~70–90° / ~250–270°) and approximately N–S (~330–20° / ~150–200°); Fig. 4). The branching, dendritic structure of valleys (Fig. 3a) indicates that the landscape may have a fluvial origin. Purely fluvial systems tend to be characterised by relatively intricate valley networks, with high stream orders, numerous tributaries, and acute tributary junction angles, whereas glacial valley networks generally have lower stream orders, fewer contributing channels, and near-orthogonal junction angles[43].

Figure 3a shows evidence for both of these types of networks, with some valleys containing complex branching structures with multiple tributaries that are suggestive of fluvial incision, while others have a simpler network structure of tributaries entering the main stem at a relatively high junction angle, more indicative of glacial incision. The valley cross profiles tend to be U-shaped, separated by sharp ridges (Fig. 3b), indicative of erosive modification by valley glaciers. It is unlikely that these valleys were incised by subglacial meltwater, as in such cases, networks tend to be non-dendritic and oriented parallel to the overall direction of ice flow[44].

The RADARSAT-derived valley networks also provide evidence for the potential coherency of the three blocks, as a number of N–S trending valleys suggest a previous connection between the blocks

(Fig. 3a). These connections have now been erased by the insertion of the troughs, so the valleys must predate the trough incision. There are also similar valley widths across each block, with median ridge-to-valley spacings of 2.1–2.7 km (Fig. 3c). Analysis of RES data (Fig. 3b) demonstrates that the mean relief (difference in elevation between ridge crests and valley floors) of the buried topography is approx. 800 m (South: 823 m; Central: 814 m; North: 757 m) with maximum relief of 1197 m (1021, 1197, and 1058 m). On average, the valley floors currently occupy similar elevations across the three blocks, with mean thalweg elevations of 238 m below sea level (South), 125 m below sea level (Central), and 206 m below sea level (North). However, valley floors are located over a range of elevations, with standard deviations of 427 m (South), 209 m (Central), and 241 m (North). Drainage density, a measure of valley length per unit area, is similar across all three blocks (0.13, 0.09, and 0.09 km per km²). The approximate uniformity of these metrics across the three blocks is indicative of a coherent single land surface.

Our analysis indicates that valley structure therefore meets criterion #1 due to its similarity across the three blocks and because it is not consistent with modern continental-scale ice flow. The dendritic valley structure, potential connectivity of valleys across the blocks, valley-ridge relief/morphology, and valley cross-sectional profiles are all consistent with a fluvial landscape subsequently modified by local-scale glaciation.

The second criterion, that the elevation of the ancient surface should be coherent when corrected for more recent glacial trough incision and ice loading, was assessed using flexural modelling. Our calculations show that the displacement induced by erosional unloading within the large-scale troughs and smaller-scale valleys has caused land surface uplift of up to 500 m (Fig. 5). This value is likely to be an upper limit because the two large troughs may have been partially developed prior to glaciation, for example, due to inherited tectonic structure and/or fluvial incision[10,33,35]. The spatial pattern of uplift is dependent on the chosen value of the effective elastic thickness of the lithosphere (Te; Fig. 5). For low Te values (e.g., 5–10 km), the flexure exhibits short wavelength warping close to the margins of the large troughs that is not consistent with the observed elevations of the intervening highlands. By contrast, the flexure computed for higher Te values of 20–40 km predicts a pattern of uplift characterised by gentle domal tilting away from the deep trough between the southern and central blocks, within which the largest eroded thicknesses are predicted (Fig. 5). This pattern of tilting is more consistent with the observed peak elevations. Higher values of Te up to 80 km or more[45] cannot be precluded, given that the sensitivity of the flexure to Te at the wavelengths of interest (up to 300 km) drops off at high Te values[46].

The flexural modelling results are consistent with the interpretation that a coherent highland landscape existed prior to the incision of the two deep troughs/fjords and that the three blocks were separated, uplifted, and subtly upwarped/domed as a result of this incision (thereby meeting criterion 2). The computed maximum amplitude of the erosion-driven flexure is ~500 m, whereas the elevations of the highland peaks are up to 1500–1700 m above sea level under ice-free conditions (Figs. 4 and 5). This implies that the highland peaks were situated at least 1–1.2 km above sea level prior to glacial incision, assuming a lack of vertical displacement driven by dynamic topography processes, which remain unconstrained in this region. The similarity of the hypsometries of the three blocks when isostatically corrected for ice loading, with consistently located modal peaks for the valley floors and ridge tops (Fig. 4), also offers further support for criterion 1.

The third criterion, that the basal thermal regime must have been cold-based to preserve the landscape, was tested by comparing our landscape to outcomes from previous numerical models of basal melting and with modern ice flow patterns. Modern ice flow is slow

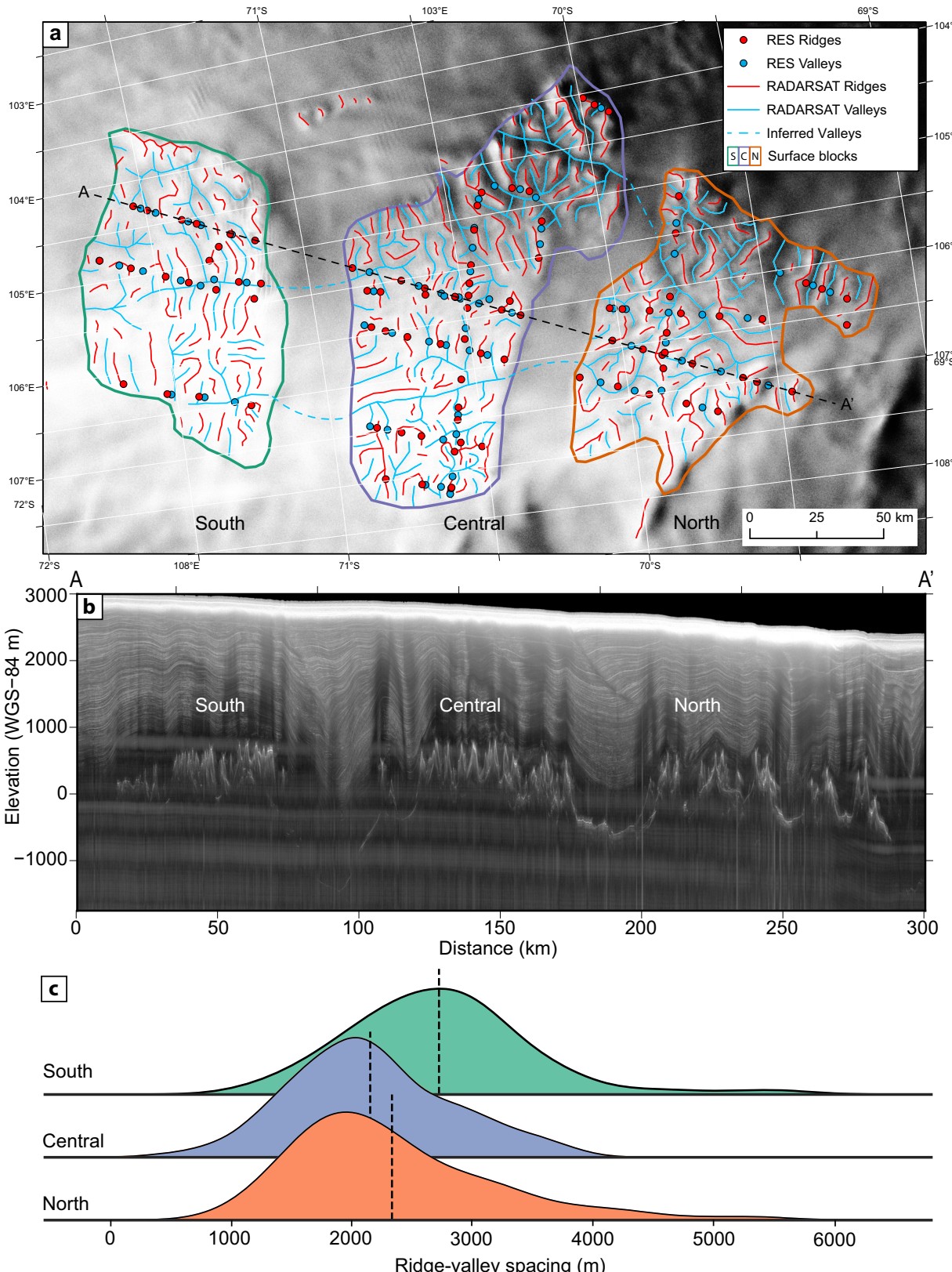

**Fig. 3 | Mapping the subglacial landscape blocks in the Highland A region reveals a coherent valley network spanning three blocks. a** Valley and ridge planforms (lines) identified in ice surface imagery compared with locations of topographic 'highs' (ridges) and 'lows' (valleys) identified from RES survey data (points). The background shows the RADARSAT ice surface imagery[42]. Dashed lines denote inferred former valley connections across south (S), central (C) and northern (N) topographic blocks. Dashed line A–A' shows the location of the radargram in panel (**b**). The map is projected in EPSG:3031 and rotated by 25 degrees and its location is shown in Fig. 2a. **b** Depth corrected 2D focused radargram from the January 2008 ICECAP campaign confirms surface features correspond to peaks and valleys. Depths relative to WGS84 ellipsoid. **c** Kernel density estimation plot of the ridge-to-valley spacing based on mapping in A. Vertical dashed lines denote means. Valley widths would be double the ridge-valley spacing. Source data are provided as a Source Data File.

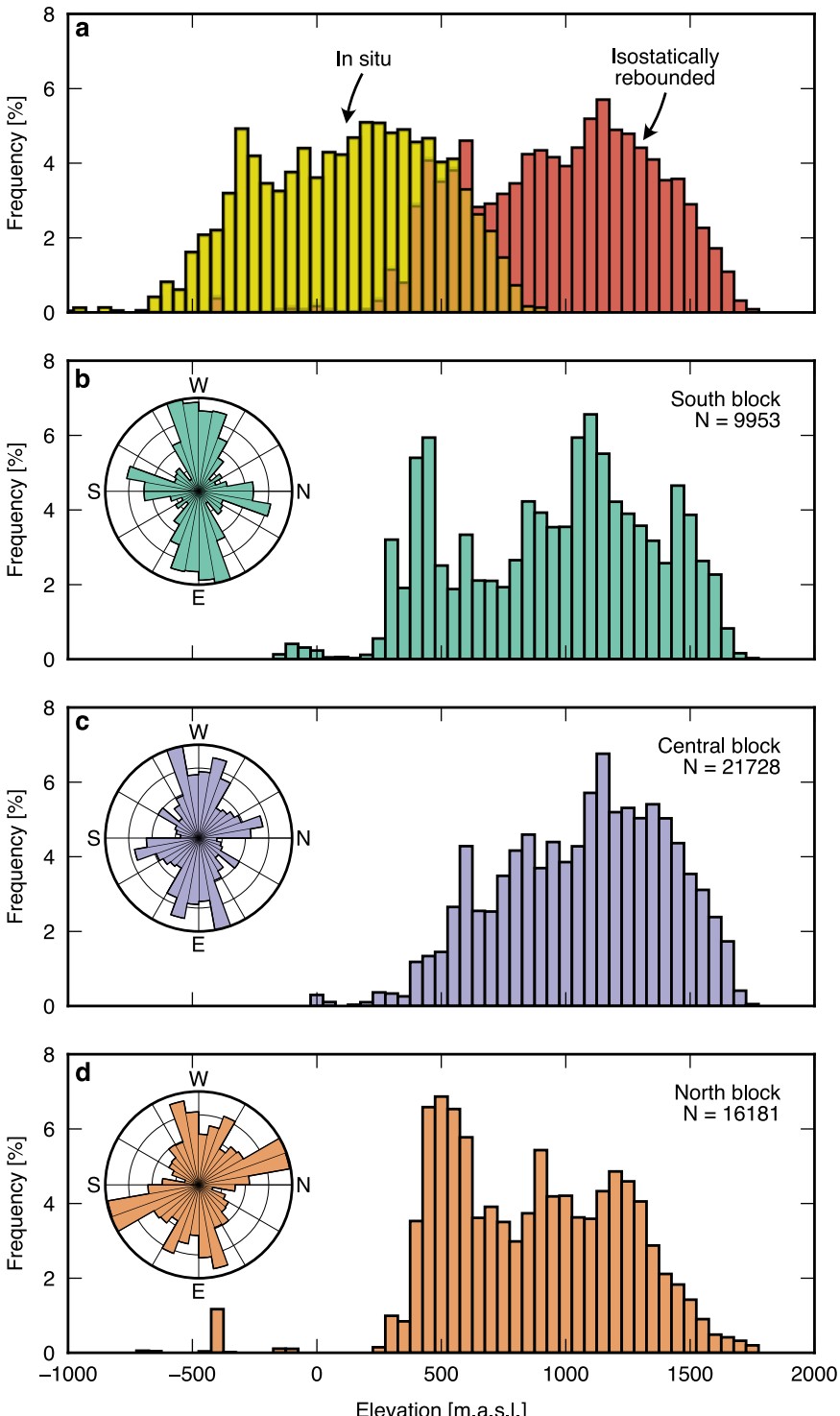

**Fig. 4 | Highland A blocks have similar hypsometries. a** Combined hypsometry for all three highland blocks under in situ (i.e., ice loaded; yellow) and isostatically rebounded (i.e., ice-free; red) conditions using a laterally heterogeneous Te model[45]. Lower panels show rebounded hypsometry for the individual landscape blocks; **b** south block, **c** central block, **d** north block. Blocks are delineated by the outlines shown in Fig. 3a. All elevations were obtained from the airborne RES data crossing the landscape blocks and are relative to present-day global mean sea level. N is the number of RES-derived bed elevation measurements within each block. Insets show polar histograms (sector diagrams) for the orientations of the valleys in each highland block. Valleys were assigned a geographic orientation but not directionality, hence the sector diagrams are symmetrical. The sector diagrams have been rotated to align with the approximate orientation of the map view in Fig. 3a. Source data are provided as a Source Data file.

(<5 m yr⁻¹), flows in a northwards direction across the blocks and does not appear to be topographically steered by the local valleys in our mapped landscape (Fig. 2e). Over the wider region, where the ice thickness is greater (~4000 m) (Fig. 2b), elevated ice flow (10−50 ma⁻¹)

at the onset zones of the Denman and Totten Glaciers is 'steered' around the highland blocks, with basal melting potentially occurring in the floors of the dissecting fjords[47] (Fig. 2f). Therefore, the scale, orientation and local relief of the upland valleys are consistent with a

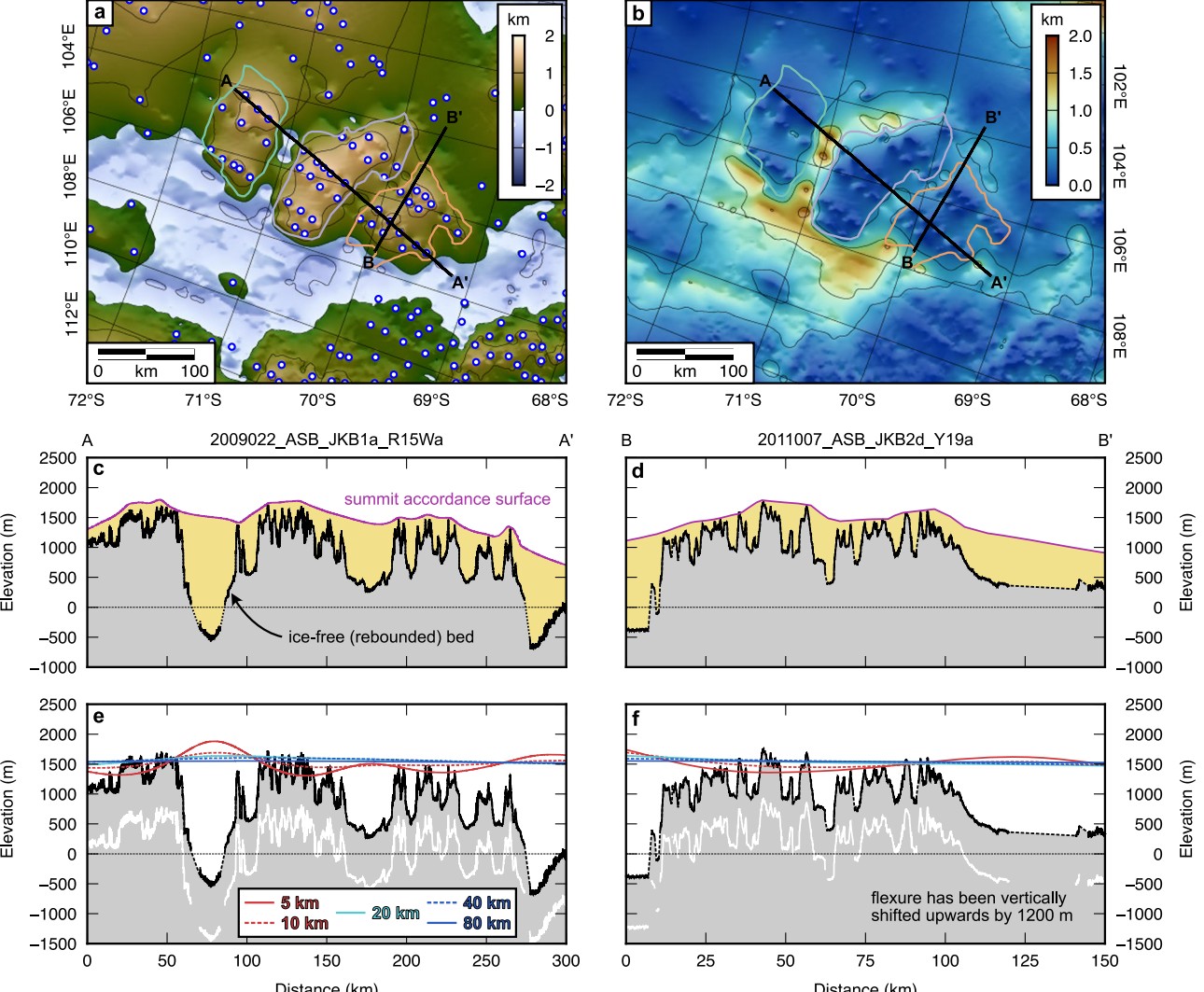

**Fig. 5 | Flexural modelling of erosion-driven uplift indicates the coherency of the three highland blocks. a** BedMachine v.3 bed topography[64] isostatically adjusted for the removal of the modern ice sheet load using a heterogeneous *Te* model[45,70]. White and blue circles mark topographic peaks used to estimate the distribution of valley and trough erosion. **b** Estimated erosion distribution, calculated by differencing the ice-free topography from a smooth accordant surface interpolated between the peaks shown in panel (**a**). **c** Erosion estimate along profile A–A′. The thickness of eroded material is shaded in yellow. Note that dashed lines indicate regions of the bed that are not imaged by RES data. **d** Same as panel (**c**) but for profile B–B′. Profile locations are shown in panels (**a**) and (**b**). **e** Computed flexural response to erosional unloading along profile A–A′ for five *Te* values. To aid visual comparison between the calculated flexure patterns and the tilt of the highland surfaces, the flexure has been vertically shifted upwards by 1200 m. The white line shows in situ bed elevation beneath the modern East Antarctic Ice Sheet. **f** Same as panel (**e**) but for profile B–B′. All elevations are relative to present-day global mean sea level.

cold basal ice temperature at the present day, and indeed since the incision of the local valleys within each block. A final piece of potential evidence for a cold basal thermal regime at present is the indication of internal reflection horizons that may indicate older ice is present over the southern block (Fig. 3b) and suggesting that basal melting must therefore be minimal otherwise, this ice would not have survived.

## Discussion

The three criteria tested by our analysis have been met, confirming the existence of a relic landscape in central East Antarctica. We now consider the sequence of events and possible age of this landscape before addressing the implications for ice sheet history.

We propose that the following relative sequence of events is required to modify an original large-scale fluvial landscape to become a set of three remnant upland blocks separated by deep troughs. This landscape evolution scenario (Fig. 6) requires interactions between tectonic structures and climate in order to dissect the original fluvial

drainage network, modify it with local ice, and subsequently preserve it beneath the continental-scale EAIS.

- The original landscape (Fig. 6a) likely developed under pre-glacial conditions before, during and after the break-up of Gondwana, when fluvial incision would have been the main agent of landscape evolution on a continental scale, with deep weathering occurring in warmer, more humid areas[48]. It is possible that the preserved dendritic valleys originally formed part of a palaeofluvial drainage pathway that radiated from the interior highlands of East Antarctica (e.g., the Gamburtsev Mountains) to the Wilkes Land coast and potentially across to Australia prior to continental separation[49]. We expect this fluvial landscape to have been characterised by relatively low local relief (<800 m) and peak elevations approximately 1 km above sea level.

- Jurassic–Cretaceous Gondwana break-up likely caused the reactivation of pre-existing (Late Palaeozoic–Triassic) rift

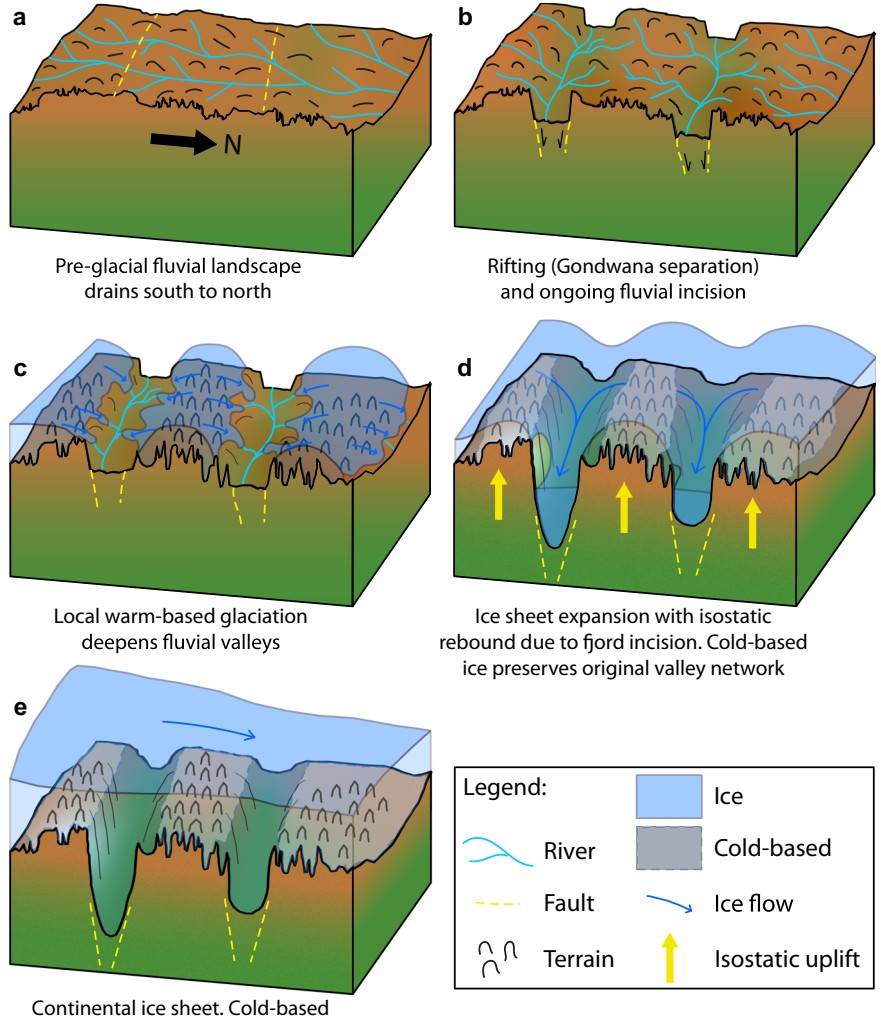

Fig. 6 | Inferred evolution of the ancient landscape. a Original fluvial land surface; b Gondwana break-up activates pre-existing tectonic structures, causing drainage re-organisation; c Local warm-based ice forms on the uplands and incises local-scale glacial valleys; d Ice expansion towards continental-scale ice sheet with cold-

based ice rapidly developed over highlands, and warm-based ice steering through and eroding the intervening structurally-controlled valleys so they become fjords; e Continental-scale ice with cold-based ice on the highlands and continental-scale ice flowing slowly over the region, largely ignoring topography.

structures in East Antarctica, as well as uplift of the Australo-Antarctic passive margin[49,50]. We suggest that faults between the three highland blocks may have been activated at this time (Fig. 6b), with associated extensional/transtensional motion separating the highlands and thus breaking the inferred cross-block valley connections (Fig. 3a). The similarity in the way the nearby Highland B region is separated into multiple blocks[37] suggests it may also have been affected by tectonic activity at this time. We infer that separation of the blocks must have occurred prior to local glaciation since otherwise, local ice would likely not have remained topographically steered[51] due to the broad uniformity in elevation/relief across the region and would instead likely have formed a broader domed icecap.

- Initial glaciation of the landscape blocks (Fig. 6c) likely saw localised warm-based valley glaciers that were steered by the inherited dendritic topography[10,51] to deepen the valleys to ~800 m of relief and generate U-shaped cross-sectional profiles. Peaks were likely preserved beneath cold-based ice and experienced isostatic uplift due to erosional unloading within the valleys (Fig. 5). This local glaciation, which may have been cyclical and paced by orbital forcing, likely modified the

landscape over timescales on the order of $10^6$ years. However, the duration and/or erosivity of this glaciation were unable to cause incision below sea level (Fig. 4).

- The ice sheet rapidly expanded to a continental scale (Fig. 6d, e). During growth, the ice sheet likely exploited the existing tectonic structures and topographic lows separating the three highlands (Fig. 6d), further reinforcing (via glacial erosion) the isolation of the three pieces of the original land surface. Preferential steering of warm-based ice through the troughs enabled a cold-based thermal regime to become established across the entirety of the adjacent highland land surfaces[52,53] (i.e., not just on the peaks but also in the upland valley floors), thus enabling widespread preservation of the landscape whilst the ice in the deeper troughs and adjacent basins was contemporaneously warm-based.

- Subsequent ice margin fluctuations may have occurred but none of these was sufficient to significantly alter the basal thermal regime or ice flow pattern over the highland blocks. Glaciological conditions amenable to landscape preservation therefore persisted.

Our proposed landscape evolution scenario indicates that the most recent erosive modification of the three highland blocks was by local-scale glaciation. Since this modification must have occurred at a

time of much-reduced EAIS extent relative to the present day, a key question for EAIS history is: when did this local-scale glacial erosion last occur? Here, we evaluate the likelihood of the landscape of local glaciation being older than four key 'thresholds' of geological age pertinent to EAIS evolution. We consider that the land surface has a likely age of at least 14 Ma based on the following logic:

First, local-scale glacial modification is not occurring contemporarily because present-day continental-scale ice flow speed, direction[41] (Fig. 2e), and locally frozen basal conditions[47] (Fig. 2f) are inconsistent with the development of the valley-scale glacial geomorphology (Fig. 3).

Second, the landscape is most likely older than 1.5 Ma. This is because numerical modelling indicates the present-day cold basal state of this part of the EAIS is likely to have persisted since at least 1.5 Ma, even allowing for greater ice thickness at the Last Glacial Maximum and a range of geothermal heat fluxes[54]. Indeed, van Liefferinge et al.[54] find that the only way to induce melting over the past 1.5 Ma would be to increase the geothermal heat flux to above 75 mW/m², a significant shift from predicted modern rates of between 30 and 60 mW/m²[55]. Furthermore, modelling of Pleistocene interglacials does not indicate a significant response of this sector of the EAIS during those times[56–58]. The landscape therefore most likely pre-dates significant Pleistocene interglacials such as MIS 5e, 11c, and 31.

Third, the landscape is likely older than ca. 14 Ma. The mPWP (ca. 3.26–3.02 Ma) is likely the warmest (highest $CO_2$ and with orbital parameters favourable to deglaciation[59]) period since ca. 14 Ma and is therefore the most likely candidate for significant retreat of the EAIS via the Denman and Totten Glaciers, which would be necessary for modification of the landscape by local ice flow. However, ice sheet model intercomparison indicates that most models do not simulate such retreat in the Aurora and Schmidt Subglacial Basins under peak mPWP climate conditions[60] unless marine ice cliff instability is invoked[25]. We therefore suggest that these surfaces predate the EAIS switch to cold polar conditions at ca. 14 Ma[17].

Finally, there remains a possibility that the landscape is older than ca. 34 Ma. This is because after East Antarctica was initially glaciated following the EOT, the extent and chronology of fluctuations of the ice margin are uncertain[11,14]. However, there is evidence from the Aurora subglacial basin that ice was reaching the continental shelf prior to the EOT[29]. Thus, while the landscape may have formed under restricted EAIS conditions between 34 and 14 Ma, it may alternatively (or additionally) reflect glacial conditions immediately prior to the establishment of the first continental-scale ice sheet at the EOT[2,9,10]. Therefore, on the basis of this evaluation of the existing understanding of Antarctic Cenozoic glacial history, we conclude that the Highland A land surface was last incised by local valley glaciers at least prior to ca. 14 Ma and possibly immediately prior to the initial growth of continental-scale ice at 34 Ma.

The presence of this preserved landscape has implications for understanding long-term EAIS behaviour. As previously shown in other regions of Antarctica[9,31,32,34,35,61], ancient landscapes can survive beneath the EAIS for periods of at least several million years and likely much longer. In order for this to be possible, the transition between two glacial states—(1) local-scale glaciation with warm-based erosive glaciers cutting U-shaped valleys and (2) a continental-scale ice sheet which is locally cold-based and preserves the landscape of local glaciation—must occur rapidly. Previous modelling of ice sheet growth has demonstrated rapid ice expansion and a snap-change in basal thermal regime from warm- to cold-based occurred in the Gamburtsev Mountains region at the EOT[10], meaning they were eroded very little[62] and we propose a similar mechanism to have occurred across Highland A. Such rapid changes in ice extent and dynamics are supported by recent ice sheet-climate modelling[63], where over a period of a few million years, either side of the EOT

simulated ice volumes oscillate between local- and continental-scale in a quasi-binary and geologically near-instantaneous manner. During the growth of the EAIS, it is therefore possible that ice existed in a relatively small local form capable of eroding the Highland A land surface or at a continental scale that preserved the landscape of local glaciation, with little time in between.

The preserved landscape we identify is somewhat unique given that it lies on the periphery of one of the 'great basins' of East Antarctica, which are of substantial interest as potential locations of past/future margin retreat via marine ice sheet instability[56,64]. The survival of the relic landscape implies that there is long-term thermal stability over these blocks of ancient terrain, with cold-based ice being the average glacial condition since landscape formation. If the ice margin were to have retreated substantially into the Aurora and Schmidt Subglacial Basins during significant past warm intervals such as the mPWP, independent ice masses remaining on the highland blocks would likely have been drained radially by outlet glaciers. We do not see evidence of radial drainage, but instead a more complex dendritic pattern and therefore deem it likely that the ice margin has never retreated far enough to enable this region to become an independent ice centre. If our interpretation is correct, this landscape may have survived because any retreat of this sector of the EAIS during past warm periods did not extend as far inland as these highland blocks, which are ~300 km inland of the modern-day grounding line. We note, however, that this landscape is likely not sensitive to, and therefore does not preclude, fluctuations within 300 km of the modern ice margin[33].

Our analysis shows that local-scale glaciation modified a pre-existing fluvial landscape. However, it is not clear whether this would have occurred under the single initial state of early ice growth at or before the EOT or whether this reflects average glacial conditions throughout the Oligocene and early Miocene prior to the onset of full polar ice sheet conditions at ca. 14 Ma[17]. Sampling of subglacial bedrock and potentially subglacial sediment would help better constrain the landscape age and we propose that given this region may not have been subject to significant erosion or basal transport for millions of years, it may be a promising area to recover a deep ice and subglacial sediment core in order to understand glacial conditions during those times. In addition, a more detailed local RES survey would help resolve the elevation and geometry of potential cirque floors[2,31], which could then be used to constrain the climate that supported local-scale glaciers on the landscape. Such a survey could be directed by our ice surface feature mapping (Fig. 3a) and the hints of cirques visible in the existing RES survey data (Fig. 3b).

Finally, given this discovery of an ancient landscape hidden in plain sight, and that of others[9,32,40,61], we propose that there will be other similar, as yet undiscovered, ancient landscapes beneath the EAIS. Systematic mapping of these landscapes, along with sampling via direct bed access, may provide significant new insights into the growth and past fluctuations of the EAIS. They may also facilitate a greater understanding of the long-term landscape evolution and tectonic history of interior East Antarctica, which is largely inferred from potential field data with little direct supporting evidence.

In conclusion, we use satellite remote sensing of ice surface morphology and imaging of the ice-bed interface from RES surveying to map a previously undiscovered landscape in the Aurora-Schmidt basins in the upper Denman and Totten glacier catchments of East Antarctica. We find that the landscape contains a valley network with a dendritic (i.e., fluvial) planform and was once a continuous land surface that has since been dissected by deep glacial troughs that may inherit earlier tectonic weaknesses. Cross profiles of the terrain from RES data indicate that this fluvial landscape has been modified by local-scale glaciers and has not been modified by ice flowing at a continental scale. The age of the land surface is uncertain, but it is likely to predate the mid-Miocene (ca. 14 Ma) and perhaps dates from the transition

from warm to glacial conditions in Antarctica following the EOT at ca. 34 Ma. The survival of the landscape implies a long-term stable basal thermal regime and that the ice margin is unlikely to have retreated as far inland as this locality during warm periods of the last 14 Ma. The landscape is more consistent with the idea that this region of East Antarctica requires greater warming than seen in (e.g.,) the mPWP for retreat to reach this far into the Aurora-Schmidt subglacial basins.

Given that modern atmospheric $CO_2$ and temperature conditions have reached levels unprecedented since the Pliocene[65,66], we are now on course to develop atmospheric conditions similar to those that prevailed between 34 and 14 Ma (ca. 3–7 °C warmer than present), with $CO_2$ conditions reaching above 500 ppm between now and 2100 under continued fossil fuel burning[67]. Under such anthropogenic forcing and the associated global heating, our work suggests the EAIS may eventually retreat enough that local ice caps would once again exist on ancient preserved landscapes at the margin of the Aurora and Schmidt basins.

## Methods

### Geomorphological mapping

Following the approach of Ross et al.[32] and Jamieson et al.[40], we use satellite remote sensing to map the likely ridge and valley structure of the Highland A region. We use contrasts in the RADARSAT[42] and REMA[68] ice surface datasets to infer minor changes in ice surface slope, which are a consequence of buried topography. RADARSAT imagery records the brightness or intensity of the reflection of the satellite-emitted radar signal, which depends on the angle of the surface slope. Typically, subglacial valleys are positioned below darker areas (lower reflection intensity) and subglacial ridges below lighter areas (higher reflection intensity) in the RADARSAT image.

REMA is an ice surface DEM; we calculate the plan and profile curvature of the DEM to identify breaks in slope that are indicative of subglacial valleys and ridges. In some locations, the change in ice surface brightness/curvature is abrupt, for example, where potentially sharp ridges or peaks are present beneath the ice. In other areas, the contrast is significantly muted, making mapping more challenging and less certain, particularly when attempting to determine valley/ridge orientations.

Using the processed REMA and RADARSAT images, we digitise the locations of valleys and ridges where clear breaks in slope or changes in radar reflection intensity occur. Connections between valleys are inferred such that if a valley is continuous across an entire block, it is then connected to the next block if this block hosts an apparently contiguous valley on the other side of the intervening trough. Within each topographic block, the spatial statistics for the mapped valleys are calculated, including their length, the ridge-valley spacing, and the drainage density (length of valley per unit area).

The location and form of the valleys are then confirmed using the available HiCARS RES survey data[37–39]. Data were focused following the approaches described in Peters et al.[69]. Using the focussed RES lines, we automatically extract ridge and valley locations along each RES line as the points of maximum and minimum elevation within a 10 km-wide moving window, respectively. The ridge and valley locations are used to ground-truth the locations of the features mapped from the REMA and RADARSAT data. We also determine the mean relief (i.e., the vertical distance between individual ridges and valleys) within the three landscape blocks. In addition to identifying the locations of ridges and valleys, we qualitatively examine the RES lines to identify whether the valleys are U-shaped or V-shaped and thus assess whether the landscape records the erosive imprint of either local valley glaciers or rivers.

### Flexural modelling

We use flexural modelling to quantify the effect of large-scale valley incision on the elevation of the highland blocks. After isostatically adjusting bed elevations for the removal of the modern ice load[70], we construct an accordant surface between the highland peaks. Peaks are identified as high points in the landscape within a circular moving window of diameter 10 km, and a smooth surface representing the land surface in the absence of valley incision is interpolated between these peaks (Fig. 5). This summit accordance surface also extends across the two wide troughs that separate the three highland blocks and across the surrounding low-lying terrain. The distribution of erosion is, in turn, inferred to be the difference between the accordant surface and the (ice-free) bed topography[71,72].

We use a 2D elastic plate model to compute the flexure, w, induced by the removal of the eroded material of thickness h. The general 2D flexure equation is expressed as

$$\nabla^2 \left[ D(x,y)\nabla^2 w(x,y) \right] + \left( \rho_{mantle} - \rho_{infill} \right) g\, w(x,y) = \left( \rho_{load} - \rho_{displace} \right) g\, h(x,y)$$

(1)

where

$$D(x,y) = \frac{E T_e(x,y)^3}{12(1 - \nu^2)}$$

(2)

The parameter $T_e$ represents the effective elastic thicknesses (a proxy for the flexural rigidity, $D$) of the East Antarctic lithosphere. For simplicity, we assume a spatially uniform $T_e$, allowing Eq. (1) to be solved analytically using a fast Fourier transform of the erosional unload and convolution with a 2D flexural isostatic response function[46]. We assume a Young's modulus (E) of 100 GPa, a Poisson ratio ($\nu$) of 0.25, a gravitational acceleration (g) of 9.81 m s$^{-2}$ and densities of 2670 and 3330 kg m$^{-3}$ for the eroded material ($\rho_{load}$) and mantle ($\rho_{mantle}$), respectively. Note that for erosional unloading, the material infilling and displaced by the flexure is air, so $\rho_{infill}$ and $\rho_{displace}$ are set to 0 kg m$^{-3}$. We perform the calculation for a range of constant $T_e$ values to test the sensitivity of the flexural response to lithospheric rigidity and compare the observed land surface elevations to the predictions of our elastic plate model for different $T_e$ values.

## Data availability

The data used for this work is the radio-echo sounding data from the ICECAP project, which is openly accessible via the Blankenship 2017 references[38,39] (HICARS1: https://doi.org/10.5067/F5FGUT9F5089; HICARS2: https://doi.org/10.5067/9EBR2T0VXUDG). The mapping data generated in this study (Fig. 3a) are openly available as GIS shapefiles at https://doi.org/10.5281/zenodo.8159223[73]. Source data are provided with this paper—these relate to the data that underlies Figs. 3c and 4. Source data are provided with this paper.

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

## Acknowledgements

We thank Felicity McCormack for supplying modelled basal melt data. Funding for geophysical data acquisition was provided by NSF grants ANT-0733025 (D.D.B., D.A.Y.), ANT-1443690 (D.D.B., D.A.Y.), NASA grant NNX09AR52G (D.D.B., D.A.Y.), the G. Unger Vetlesen Foundation (D.D.B., D.A.Y.) and the UK NERC grant NE/D003733/1 (M.J.S.). G.J.G.P. was funded by UK NERC grant NE/L002590/1. For the purpose of open access, the author has applied a Creative Commons Attribution (CC BY) license to any Author Accepted Manuscript version arising from this submission. This is University of Texas Institute of Geophysics contribution 3963.

## Author contributions

S.S.R.J. and N.R. conceived the work. D.A.Y., D.D.B., M.J.S. and J.G. provided geophysical survey data. S.S.R.J., G.J.G.P., F.J.C., D.A.Y. and S.Y. conducted the analysis. S.S.R.J., N.R., G.J.G.P., F.J.C. and M.J.S. drafted the manuscript and S.S.R.J., N.R., G.J.G.P., F.J.C., D.A.Y., S.Y., D.D.B., J.G. and M.J.S. provided revisions to the manuscript.

## Competing interests

The authors declare no competing interests.
