## [Peer Review File · Nature Communications]

An ancient river landscape preserved beneath the East Antarctic Ice SheetReviewers' Comments:

Reviewer #1:

Remarks to the Author:

The manuscript presents results of careful geomorphological analysis of the ice bed over subglacial highlands in a key location beneath the East Antarctic Ice Sheet and describes a landscape evolution scenario that is persuasive. Satellite remote sensing data and airborne radio-echo sounding data are integrated to generate the geomorphological dataset, and lithospheric flexure modelling and inversion of aeromagnetic anomaly profiles provide additional constraints on the interpretation. The manuscript is well written and has a clear structure. The implications for long-term ice sheet history are significant, addressing a gap in knowledge (the extent of ice sheet retreat during past warm intervals) that is difficult to constrain by other means and relevant to potential future change.

I have only minor comments, which are included in a commented version of the manuscript that I am uploading with this review.

The figures are of publishable quality and the associated captions describe clearly what is shown. One of my more significant comments though is that labels on the profiles in Fig. 6 appear to indicate the orientation incorrectly, and in fact it would be better to present these profiles in the reverse orientation.

The one component of the manuscript I do not find particularly convincing is the magnetic inversion work. I am fairly confident that the observed magnetic anomalies could be matched satisfactorily by a model with a smoothly undulating top to a buried magnetic layer that does not require any fault structures, either compressional or extensional. I also find the lack of consideration of airborne gravity data, which various online sources suggest were acquired on the ICECAP surveys, in the inversion modelling puzzling. A joint inversion of magnetic and gravity data would be more convincing. Overall though, this is a secondary issue. The inference that the broad, deep valleys separating the subglacial highland blocks result from erosion along lines of tectonic weakness seems very likely.

There are a couple of points in the text I particularly want to draw attention to:

Lines 120-122 - it is stated that there is "a subtle 'domed' pattern of tilting away from the northern and central blocks, which exhibit the highest elevations". In fact Fig. 3 shows that the southern and central blocks have the highest elevations. I presume that this reflects temporary confusion about the geographic orientation of the study area maps at the time of writing?

Lines 188-189 - It should be acknowledged here that the estimate of elevation of the highland peaks prior to glacial incision assumes no significant dynamic topography effects since the landscape was formed.

R D Larter
12th June 2023

Reviewer #2:

Remarks to the Author:

I enjoyed reading this manuscript. It was well written, which translates into only a few minor edits.

I will focus on the larger picture. The authors have built a strong case that the landscape they depict through geophysical imaging through the EAIS is a fluvial landscape modified by local wet-based glaciers. This landscape has been effectively entombed by thick cold ice for a long time.

It seems to me that this is straight forward, and that the array of evidence is sufficient to make the case. I do not get much from the geophysical magnetic modeling, and think that piece could be dropped without significant loss. It would in fact make the argument less convoluted. They then talk about the likely times at which the landscape could have formed, or what constraints their entombed landscape places on the cold ice history. While this is effectively speculative, I think it is still worth while, as it forces the audience to face the longer >30Ma history of ice occupation in EA.

Minor edits: (line number, then my suggested wording; comments in [])

101 have been shown to reflect the

160 is suggestive of a coherent

171-172 [I note that the troughs, if originally developed by erosion of rock that is made inherently weak by tectonic grain or damage, could not have been eroded to below sealevel. Rivers don't do that. The overdeepened to << sealevel has to have been accomplished by glaciers, indeed temperate-based glaciers.

192-204 It is here, in this 3rd criterion and the magnetic modeling, that I am less supportive. Any "selective preservation" could occur through steering of the wet-based ice when big ice first immerses the landscape. It does not in my opinion require that geologic structures be in place to accomplish this, merely that highest order drainages will naturally have the highest ice discharges and hence slide most rapidly and hence dominate. See Kessler and Anderson for one of no doubt many modeling experiments aimed at demonstrating exactly this point.

208 does not appear to be topographically steered

215 is the indication of internal reflection horizons that may

217 3b); this suggests that

289 pre-dates Pleistocene deep interglacials such as [and perhaps add their ages]

364 and propose that it was once a continuous

[I note that there is no mention of the magnetic modeling in the conclusions, and agree with this omission – that piece of the argument is not needed to make the case.]

Figures

3. A. There is room on the figure to label South Central and North as words. It takes the audience longer to figure out that the colors of the outlines correspond to these.

I like the use of the same colors on the spacing distributions in C.

References the authors may want to check out

Kessler, M. A., Anderson, R. S. and Briner, J. P., 2008, Fjord insertion into continental margins by topographic steering of ice. *Nature Geoscience* 1(6): 365-369.

Reviewer #3:

Remarks to the Author:

The reported discovery of a glacially dissected but otherwise geomorphologically-intact fluvial landscape bordering the Aurora Subglacial Basin (East Antarctica) is significant in implying rapid ice-sheet expansion after the mid-Miocene thermal maximum and perhaps also at the Eocene-Oligocene transition. I recommend publication in *Nature Communications* after very minor revision.

I recommend the addition of a small number of references, listed below by line number. I also suggest that the authors consider commenting on a current controversy concerning global average depths of glacial erosion during Cryogenian panglacial ('snowball Earth') episodes, when East Antarctic climatic conditions prevailed in the tropics as a result of high planetary albedo due to ocean-wide ice shelves ('sea glaciers'), maintained at all paleolatitudes by equatorward flow (e.g., Tziperman et al. 2012 *J. Geophys. Res.* 117, C05016, doi:10.1029/2011JC007730).

Comments by manuscript line numbers:

38-39 Wherever possible, I avoid using the terms 'greenhouse' and 'icehouse', which refer specifically to processes within enclosed spaces that are misleading when applied to climate. Such terms makes it more difficult for both students and the public to understand the basic physics behind climate change. I suppose we are stuck with 'greenhouse' but must we add 'icehouse' to the muddle. An icehouse is an enclosure that is kept cold by placing blocks of ice inside. However, the heat that the ice takes up as it warms and melts is the same heat that was given off when the ice blocks were frozen in the first place.

59 Add reference: Blackburn T, Edwards GH, Scudder M, Piccione G, Hallet B, Zachos JC, Cheney B & Babbe JT (2020) Ice retreat in Wilkes Basin of East Antarctica during warm interglacial. *Nature* 583, 554–559.

63 Add reference(s) to ANDRILL results: Naish T, Powell R & 54 co-authors (2009) Obliquity-paced Pliocene West Antarctic ice sheet oscillations. *Nature* 458, 322–328; Passchier S, Browne G, Field B, Fielding CR, Krissk LA, Panter K, Pekar SF & ANDRILL Team (2011) Early and middle Miocene Antarctic glacial history from the sedimentary facies distribution in the AND–2A drill core, Ross Sea, Antarctica. *Geol. Soc. Am. Bull.* 123(11/12), 2352–2365; Rosenblume JA & Powell RD (Glacial sequence stratigraphy of ANDRILL–1B core reveals a dynamic subpolar Antarctic Ice Sheet in Ross during the late Miocene. *Sedimentology* 66, 2072–2097.

308 Add reference(s) to results from Gamburtsev Subglacial Mountains: Cox SE, Thomson SN, Reiners PW, Hemming SR & van de Flierdt T (2010) Extremely low long-term erosion rates around the Gamburtsev Mountains in interior East Antarctica. *Geophys. Res. Lett.* 37, L22307; Creyts TT, Ferraccioli F, Bell RE, Wolovick M, Corr H, Rose KC, Frearson N, Damaske D, Jordan T, Braaten D & Finn C (2014) Freezing of ridges and water networks preserves the Gamburtsev Subglacial Mountains for millions of years. *Geophys. Res. Lett.* 41, 061491.

358 Optional added discussion point: Controversy has recently developed concerning depths of glacial erosion during prolonged glaciation of all continents during two discrete Cryogenian episodes of 56 and 6–16 Myr duration for the Sturtian and Marinoan snowball episodes, respectively. The debate resembles an earlier one concerning Quaternary glacial erosion by the Laurentide Ice Sheet (White 1972 *Geol. Soc. Am. Bull.* 48: 1037–1056; Sugden 1978 *J. Glaciol.* 20(83): 367–391). On the one hand, stratigraphic projections and global average sediment accumulation rates during the Cryogenian episodes imply average glacial erosion rates of only ~4 m/Myr assuming mass conservation (Partin & Sadler 2016 *Geology* 44: 1019–1022; Hoffman PF 2023 *Can. J. Earth Sci.* [dx.doi.org/10.1139/cjes-2022-0004](https://doi.org/10.1139/cjes-2022-0004)). In contrast, average depths of glacial erosion of 3–5 vertical kilometres (i.e., 40–66 m/Myr for both episodes combined) have been inferred from other sources of information, with the deficit in preserved mass explained by sediment subduction (Keller CB, Husson JM, Mitchell RN, Bottke WF, Gernon TM, Boehnke P, Bell EA, Swanson-0Hysell NL & Peters SE 2019 *Proc. Natl Acad. Sci. USA* 117(4): 1136–1145). Much of the argument in Keller et al. (2019) revolves around a supposed stepwise increase in sediment accumulation rate that occurred 90 Myr after the pagnlacial episodes ended, which to this reviewer is a completely non-physical association. Nevertheless, the subduction-lubrication implication has been seized upon by metamorphic geologists as facilitating the onset of 'modern-style' plate tectonics (Sobolev & Brown 2019 *Nature* 570(7759): 52–57). A Neogene Antarctic perspective would be a helpful addition to this discussion.

signed: P.F. Hoffman, 12 June 2023

We thank the 3 reviewers for their positive responses and helpful comments.

Please find below our responses (in **bold**) to each reviewer comment:

Reviewer #1 (Remarks to the Author):

The manuscript presents results of careful geomorphological analysis of the ice bed over subglacial highlands in a key location beneath the East Antarctic Ice Sheet and describes a landscape evolution scenario that is persuasive. Satellite remote sensing data and airborne radio-echo sounding data are integrated to generate the geomorphological dataset, and lithospheric flexure modelling and inversion of aeromagnetic anomaly profiles provide additional constraints on the interpretation. The manuscript is well written and has a clear structure. The implications for long-term ice sheet history are significant, addressing a gap in knowledge (the extent of ice sheet retreat during past warm intervals) that is difficult to constrain by other means and relevant to potential future change.

Thanks for your positive and careful review.

I have only minor comments, which are included in a commented version of the manuscript that I am uploading with this review. **We copy and paste those comments below so that you can identify how we respond.**

The figures are of publishable quality and the associated captions describe clearly what is shown. One of my more significant comments though is that labels on the profiles in Fig. 6 appear to indicate the orientation incorrectly, and in fact it would be better to present these profiles in the reverse orientation. **This point becomes irrelevant because we have removed the magnetic inversion (and thus figure 6) from the paper on the basis of the comments below and from reviewer 2. The associated profile locations have therefore also been removed from Figure 5.**

The one component of the manuscript I do not find particularly convincing is the magnetic inversion work. I am fairly confident that the observed magnetic anomalies could be matched satisfactorily by a model with a smoothly undulating top to a buried magnetic layer that does not require any fault structures, either compressional or extensional. I also find the lack of consideration of airborne gravity data, which various online sources suggest were acquired on the ICECAP surveys, in the inversion modelling puzzling. A joint inversion of magnetic and gravity data would be more convincing. Overall though, this is a secondary issue. The inference that the broad, deep valleys separating the subglacial highland blocks result from erosion along lines of tectonic weakness seems very likely. **Given several reviewers are critical of the magnetic inversion, and on the basis that very little of our core findings rely upon it, we have removed the magnetic inversion from the manuscript. This includes the figure, a small number of sentences in the text, and a section of the methods. Two references (Clinger et al and Talwani et al) are removed in the process, which enables us to add references in response to later reviewer comments. We retain a point about likely tectonic inheritance in our proposals for the sequence of events and boost our conclusions to mention this inheritance.**

There are a couple of points in the text I particularly want to draw attention to:

Lines 120-122 - it is stated that there is “a subtle ‘domed’ pattern of tilting away from the northern and central blocks, which exhibit the highest elevations”. In fact Fig. 3 shows that the southern and central blocks have the highest elevations. I presume that this reflects temporary confusion about

the geographic orientation of the study area maps at the time of writing? **Correct - orientations were confused as the reviewer indicates. Changed text to 'southern and central blocks'.**

Lines 188-189 – It should be acknowledged here that the estimate of elevation of the highland peaks prior to glacial incision assumes no significant dynamic topography effects since the landscape was formed. **To clarify, we have added a sentence relating to the assumption that no dynamic topographic effects are accounted for because they have never been constrained in this region.**

Reviewer #1 (comments transferred from the manuscript pdf):

Line 28: Or a single rapid transition? **We agree – it could have been singular or multiple and we therefore change the sentence 'Preservation of the relic surfaces indicates an absence of significant warm-based ice flow throughout their history, suggesting rapid transitions between restricted and expanded ice sheets' to 'Preservation of the relic surfaces indicates an absence of significant warm-based ice throughout their history, suggesting any transitions between restricted and expanded ice were rapid'.**

Line 48: See also Perez et al., 2021 and 2022 <https://doi.org/10.1130/B35814.1> <https://doi.org/10.1016/j.gloplacha.2022.103891> **These are good suggestions – we have included the 2021 Perez paper. This is balanced by the removal of the Talwani paper in the methods section related to the geophysical inversion.**

Line 51: This could be read as meaning that most of the ice expanding onto the continental shelf was cold based, which seems unlikely. To eliminate this ambiguity insert "widespread" before "cold-based" and replace "expansions to, and contractions from," by "advances and retreats of the ice-sheet margin to and from". **Done. Now reads: '...with evidence for widespread cold-based ice and advances and retreats of the ice-sheet margin to and from the continental shelf'**

Line 61: What about glacial unconformities in seismic profiles? For example, see Gulick et al. (2017) <https://doi.org/10.1038/nature25026> **Our wording already includes the statement '...and in the offshore sediments on the continental shelf' which encompasses the idea of recording unconformities so we don't change the text. However, we do add reference to the Gulick paper here, and in our 'when was the landscape last modified by local glaciation' section because that paper does identify evidence for early ice in the region. The addition of this reference is balanced by the removal of the Clinger et al reference which was a component of the geophysical inversion section.**

Line 86 (and several other places): 'landsurface' – separate words? **All instances in manuscript changed to 'land surface'.**

Line 118: Perhaps add "or approximately the same as the area as Maryland" for the benefit of US readers. **We do not make this change on the basis that we cannot provide examples for people from every different region. We retain the 'Wales' reference on the basis that it's a country and thus perhaps a more easily understandable analogue.**

Line 121: On Fig. 3B the northern block (nearest A') appears to be the one with the lowest elevation. Should this be "away from southern and central" here? **Reviewer is correct. We have changed to '...southern and central blocks'.**

Line 125: The ice flows, the surface velocity field describes the pattern of flow. Rephrase. **Removed 'surface velocity' so it now says '... modern ice flows broadly northwards...'**.

Line 154: Surely this should be 1197 m, the same as stated for the Central block? **Reviewer is correct – changed to 1197 m.**

Line 188: You should acknowledge that this is disregarding any possible dynamic topography effects since the landscape was formed. **Agreed. As mentioned above, we have added a sentence to indicate dynamic topography is not accounted for because it is not constrained in the region.**

Line 194: Were gravity data collected along the survey lines? If so, why were these not used to carry out a joint inversion? **Gravity data were collected. However, this point becomes irrelevant, as mentioned above, because we remove the magnetic inversion component of the manuscript.**

Line 215: change 'are' to 'is' **No change. We are talking about 3 measures (scale, orientation and relief) and therefore this should remain plural.**

Line 276: A personal view, but I have an aversion to the use of "believe" in scientific papers. How about "consider"? **This is a useful change – now says 'consider'.**

Line 739 (Fig 1 caption): The MMCO and mid-Piacenzian warm period are intervals rather than transitions. Transitions are what followed them (the MMCT and intensification of glaciations at the start of the Pleistocene, respectively). **Given we show both a climate transition and some climate intervals we change the wording to: '...timings of major climate intervals or transitions...'**

Line 739 (Fig1): The "ice sheets come and go" label gives the impression that the ice sheets were entirely lost during some warm intervals. As far as I am aware nobody is proposing that. Perhaps the meaning here can be expressed more clearly with a modified phrase? **We have changed the blue text in the figure to say 'ice sheets wax and wane'.**

Line 800 (Fig 6): In Figs 5A and 5B C' and D' are shown as being at the northern end of the two profiles, but in this figure C' and D' are shown as being at the "South block" end of the profiles. Based on the relative depths of the two main troughs it appears that the profiles in this figure are displayed with their southern end to the right, in which case the C, C' and D, D' labeling is the wrong way round. However, assuming that this inference is correct, I think it would be better to reverse the orientation of the profiles in this figure so that they are consistent with the orientation that A-A' is displayed in Figs 3 and 5. **This comment is no longer relevant given our removal of the magnetic inversion modelling (see above).**

R D Larter
12th June 2023

Reviewer #2 (Remarks to the Author):

I enjoyed reading this manuscript. It was well written, which translates into only a few minor edits. **Thank you for your positive review.**

I will focus on the larger picture. The authors have built a strong case that the landscape they depict through geophysical imaging through the EAIS is a fluvial landscape modified by local wet-based

glaciers. This landscape has been effectively entombed by thick cold ice for a long time.

It seems to me that this is straight forward, and that the array of evidence is sufficient to make the case. I do not get much from the geophysical magnetic modeling, and think that piece could be dropped without significant loss. It would in fact make the argument less convoluted. They then talk about the likely times at which the landscape could have formed, or what constraints their entombed landscape places on the cold ice history. While this is effectively speculative, I think it is still worth while, as it forces the audience to face the longer >30Ma history of ice occupation in EA. **Agreed – we remove the magnetic inversion modelling (see also reviewer 1) to simplify the argument without weakening it.**

Minor edits: (line number, then my suggested wording; comments in [])

101 have been shown to reflect the. **Agree – changed to ‘...which have previously been shown to reflect the large-scale subglacial topography...’.**

160 is suggestive of a coherent **We retain ‘indicative’ as a more certain word than ‘suggestive’ – we think the evidence allows for this certainty.**

171-172 [I note that the troughs, if originally developed by erosion of rock that is made inherently weak by tectonic grain or damage, could not have been eroded to below sealevel. Rivers don’t do that. The overdeepened to << sealevel has to have been accomplished by glaciers, indeed temperate-based glaciers. **Yes indeed, and sea level was different in the past. Which is why our wording already contains the phrase ‘...the two large troughs may have been partially developed prior to glaciation...due to... fluvial incision.’ – in other words, that partial development is by rivers and then glaciers subsequently exploit and overdeepen those valleys. No change made.**

192-204 It is here, in this 3rd criterion and the magnetic modeling, that I am less supportive. Any “selective preservation” could occur through steering of the wet-based ice when big ice first immerses the landscape. It does not in my opinion require that geologic structures be in place to accomplish this, merely that highest order drainages will naturally have the highest ice discharges and hence slide most rapidly and hence dominate. See Kessler and Anderson for one of no doubt many modeling experiments aimed at demonstrating exactly this point. **Agreed – the magnetic inversion modelling has been removed throughout. We already refer to the point about topographic steering (Kessler) later and in our section on the possible sequence of events we now make a point that there is nevertheless a possibility that such large-scale drainage is pre-conditioned by tectonic structure.**

208 does not appear to be topographically steered **Added ‘to be’.**

215 is the indication of internal reflection horizons that may **Changed to ‘...is the indication of internal reflection horizons that may...’.**

217 3b); this suggests that **No change – we feel our wording flows better and the suggestion was not one that would alter meaning in any way.**

289 pre-dates Pleistocene deep interglacials such as [and perhaps add their ages] **Changed to ‘...pre-dates significant Pleistocene interglacials...’**. **We do not add ages because we already list the marine isotope stages and each of these has an age range so the text would get more convoluted – the stage numbers are indicator enough of timing to those who care.**

364 and propose that it was once a continuous **No change. The suggested wording (‘propose’) is less certain and we feel that the evidence is enough to be certain about this without needing to ‘propose’ it. It simply ‘is’.**

[I note that there is no mention of the magnetic modeling in the conclusions, and agree with this omission – that piece of the argument is not needed to make the case.] **Agreed – and we have removed it from other areas of the paper.**

Figures

3. A. There is room on the figure to label South Central and North as words. It takes the audience longer to figure out that the colors of the outlines correspond to these.

I like the use of the same colors on the spacing distributions in C. **We have added the labels ‘South’, ‘Central’ and ‘North’ to panels a and b in figure 3.**

References the authors may want to check out

Kessler, M. A., Anderson, R. S. and Briner, J. P., 2008, Fjord insertion into continental margins by topographic steering of ice. *Nature Geoscience* 1(6): 365-369. **Thanks. We already cite this paper in the manuscript in relation to the importance of large-scale topography steering ice flow. We cite this paper in a few additional locations in terms of discussing topographic steering.**

Reviewer #3 (Remarks to the Author):

The reported discovery of a glacially dissected but otherwise geomorphologically-intact fluvial landscape bordering the Aurora Subglacial Basin (East Antarctica) is significant in implying rapid ice-sheet expansion after the mid-Miocene thermal maximum and perhaps also at the Eocene-Oligocene transition. I recommend publication in *Nature Communications* after very minor revision. **Thank you for your positive review.**

I recommend the addition of a small number of references, listed below by line number. I also suggest that the authors consider commenting on a current controversy concerning global average depths of glacial erosion during Cryogenian panglacial (‘snowball Earth’) episodes, when East Antarctic climatic conditions prevailed in the tropics as a result of high planetary albedo due to ocean-wide ice shelves (‘sea glaciers’), maintained at all paleolatitudes by equatorward flow (e.g., Tziperman et al. 2012 *J. Geophys. Res.* 117, C05016, doi:10.1029/2011JC007730). **We have decided that given the restriction on the number of references within the paper, we do not have space to refer to a period of time to which the manuscript does not point directly.**

Comments by manuscript line numbers:

38-39 Wherever possible, I avoid using the terms ‘greenhouse’ and ‘icehouse’, which refer

specifically to processes within enclosed spaces that are misleading when applied to climate. Such terms makes it more difficult for both students and the public to understand the basic physics behind climate change. I suppose we are stuck with 'greenhouse' but must we add 'icehouse' to the muddle. An icehouse is an enclosure that is kept cold by placing blocks of ice inside. However, the heat that the ice takes up as it warms and melts is the same heat that was given off when the ice blocks were frozen in the first place. **We have changed the various mentions (lines 38-39, 229, 299, and 368) of 'icehouse' (and 'greenhouse') and have rewritten the sentences to simply reflect there is a climate transition which leads to Antarctica being intensively glaciated. We refer to the Eocene-Oligocene Transition (EOT) in this respect, and step back from the use of the less formal terminology.**

59 Add reference: Blackburn T, Edwards GH, Scudder M, Piccione G, Hallet B, Zachos JC, Cheney B & Babbe JT (2020) Ice retreat in Wilkes Basin of East Antarctica during warm interglacial. *Nature* 583, 554–559. **We do not currently have space to add an additional reference in this respect - there are already a few on this aspect and we unfortunately cannot cite all papers that indicate fluctuations occur.**

63 Add reference(s) to ANDRILL results: Naish T, Powell R & 54 co-authors (2009) Obliquity-paced Pliocene West Antarctic ice sheet oscillations. *Nature* 458, 322–328; Passchier S, Browne G, Field B, Fielding CR, Krissk LA, Panter K, Pekar SF & ANDRILL Team (2011) Early and middle Miocene Antarctic glacial history from the sedimentary facies distribution in the AND-2A drill core, Ross Sea, Antarctica. *Geol. Soc. Am. Bull.* 123(11/12), 2352–2365; Rosenblume JA & Powell RD (Glacial sequence stratigraphy of ANDRILL-1B core reveals a dynamic subpolar Antarctic Ice Sheet in Ross during the late Miocene. *Sedimentology* 66, 2072–2097. **Given referencing limits, we cannot refer to all of these. We already refer to Passchier et al (2011) and refer to an earlier paper by Naish et al (2001) on orbitally paced oscillations of the ice. Given the restriction on number of references we can use, we make no change here.**

308 Add reference(s) to results from Gamburtsev Subglacial Mountains: Cox SE, Thomson SN, Reiners PW, Hemming SR & van de Flierdt T (2010) Extremely low long-term erosion rates around the Gamburtsev Mountains in interior East Antarctica. *Geophys. Res. Lett.* 37, L22307; Creyts TT, Ferraccioli F, Bell RE, Wolovick M, Corr H, Rose KC, Frearson N, Damaske D, Jordan T, Braaten D & Finn C (2014) Freezing of ridges and water networks preserves the Gamburtsev Subglacial Mountains for millions of years. *Geophys. Res. Lett.* 41, 061491. **We already refer to a key paper on the Gamburtsevs (Rose et al., 2013) and the Cox paper does not make a different point in terms of the survival of the landscape (which is the point being made here). The Rose paper is more relevant given it is an exploration of a new map of the topography whereas the Cox paper is an analysis of the product of erosion and does not analyse the topography itself. The Creyts paper is a more controversial one in terms of whether the supercooling adds extra preservation above and beyond simply having a cold-based ice sheet. Therefore, we do not add reference to either of these papers given that we are already limited in the number of papers we can refer to.**

358 Optional added discussion point: Controversy has recently developed concerning depths of glacial erosion during prolonged glaciation of all continents during two discrete Cryogenian episodes of 56 and 6–16 Myr duration for the Sturtian and Marinoan snowball episodes, respectively. The debate resembles an earlier one concerning Quaternary glacial erosion by the Laurentide Ice Sheet (White 1972 *Geol. Soc. Am. Bull.* 48: 1037–1056; Sugden 1978 *J. Glaciol.* 20(83): 367–391). On the one hand, stratigraphic projections and global average sediment accumulation rates during the

Cryogenian episodes imply average glacial erosion rates of only ~4 m/Myr assuming mass conservation (Partin & Sadler 2016 *Geology* 44: 1019–1022; Hoffman PF 2023 *Can. J. Earth Sci.* dx.doi.org/10.1139/cjes-2022-0004). In contrast, average depths of glacial erosion of 3–5 vertical kilometres (i.e., 40–66 m/Myr for both episodes combined) have been inferred from other sources of information, with the deficit in preserved mass explained by sediment subduction (Keller CB, Husson JM, Mitchell RN, Bottke WF, Gernon TM, Boehnke P, Bell EA, Swanson-OHysell NL & Peters SE 2019 *Proc. Natl Acad. Sci. USA* 117(4): 1136–1145). Much of the argument in Keller et al. (2019) revolves around a supposed stepwise increase in sediment accumulation rate that occurred 90 Myr after the pagnlacial episodes ended, which to this reviewer is a completely non-physical association. Nevertheless, the subduction-lubrication implication has been seized upon by metamorphic geologists as facilitating the onset of ‘modern-style’ plate tectonics (Sobolev & Brown 2019 *Nature* 570(7759): 52–57). A Neogene Antarctic perspective would be a helpful addition to this discussion.

We can see the similarities/links between our paper and work which seeks to understand Cryogenian erosion patterns. However, we do not see the need to bring the Cryogenian evidence into our investigation because we already have sufficient evidence to make our conclusions (as the other reviewers both confirm). Given the reviewer indicates this would be optional, no change is made. Indeed, making such a change would require a large number of additional references for which there is no space.

signed: P.F. Hoffman, 12 June 2023

Additional changes:

We have put the valley and ridge map data in an online open access repository (zenodo) as per the journals repository suggestions, and point the reader to this in the Data Availability section of the paper.

We have added a ‘source data’ file as a supplementary dataset attached to the journal and this contains the data that underlies the plotting of the valley spacing in Figure 3c, and the plotting of the hypsometries and valley orientations in figure 4. We have added the statement about the inclusion of source data in the Data Availability section of the paper.

Reviewers' Comments:

Reviewer #1:

Remarks to the Author:

The revisions comprehensively address all my comments.

I am pleased to see that a decision was taken to remove the magnetic inversion from the paper. After submitting my review I thought that perhaps I should have made a clear recommendation to do that as I don't think its conclusions are robust, but more importantly it was not critical to the paper.

I noticed rebuttal of one minor grammatical point I made, but the rebuttal relates to the previous sentence to the one I suggested the change in and the change I recommended does actually seem to have been made.

Reviewer #2:

Remarks to the Author:

The authors have done a good job of responding to all three reviews, it seems. I am perfectly satisfied with the current manuscript, and recommend its acceptance for publication.